# A Multicentre Italian Study on the Psychological Impact of an Inconclusive Cystic Fibrosis Diagnosis after Positive Neonatal Screening

**DOI:** 10.3390/children10020177

**Published:** 2023-01-18

**Authors:** Antonella Tosco, Diletta Marino, Sara Polizzi, Valentina Tradati, Rita Padoan, Claudia Giust, Benedetta Fabrizzi, Giovanni Taccetti, Lucia Merli, Vito Terlizzi

**Affiliations:** 1Paediatric Unit, Cystic Fibrosis Regional Reference Center, Department of Translational Medical Sciences, University of Naples Federico II, 80131 Naples, Italy; 2Freelance Psychologist, 50139 Florence, Italy; 3Cystic Fibrosis Regional Support Center, University of Brescia, ASST Spedali Civili of Brescia, 25123 Brescia, Italy; 4Scientific Board Italian CF Registry, 00100 Rome, Italy; 5Cystic Fibrosis Regional Reference Center, Mother-Child Department, United Hospitals, 60131 Ancona, Italy; 6Meyer Children’s Hospital IRCCS, Cystic Fibrosis Regional Reference Center, Department of Paediatric Medicine, Viale Gaetano Pieraccini 24, 50139 Florence, Italy; 7Azienda Sanitaria Toscana Centro, Palliative Care Unit, 50100 Florence, Italy

**Keywords:** newborn screening, Cystic Fibrosis transmembrane conductance regulator-related metabolic syndrome (CRMS), Cystic Fibrosis screen-positive inconclusive diagnosis (CFSPID), psychological impact

## Abstract

Background: An inconclusive diagnosis of cystic fibrosis (CF) after positive newborn screening (NBS) may cause parental distress. We compared the psychological impact of CF transmembrane conductance regulator-related metabolic syndrome (CRMS)/CF screen-positive, inconclusive diagnosis (CFSPID), and clear CF diagnosis, on parents. Methods: The participants were administered the Generalized Anxiety Disorder Scale, Patient Health Questionnaire-9, and the Italian version of the Impact of Event Scale-Revised as quantitative tools and semi-structured interviews as qualitative tools. Parental experience, child representation, relationships, future information, and perception of health status were investigated. Interviews were recorded and transcribed verbatim maintaining anonymity. Results: Thirty-two families were enrolled: sixteen with CF and CRMS/CFSPID, respectively. Anxiety and depression values were high in both groups, as were the measurement of traumatic impact subscales: avoidance, intrusiveness, and hyperarousal. The children’s health was evaluated by respective parents as being nearly healthy. Conclusions: Our results highlight negative psychological impacts, including emotional and affective representations, on parents of children with inconclusive CF diagnosis compared with those with clear diagnosis.

## 1. Introduction

Cystic fibrosis (CF) is the most common life-shortening inherited disease in Caucasians and is caused by variants in the CF transmembrane conductance regulator (*CFTR*) gene. The CF phenotype is characterised by chronic lung disease and exocrine pancreatic insufficiency; it is associated with nutrient malabsorption, contributing to undernutrition, impaired growth, hepatobiliary manifestations, and male infertility [1]. The introduction and widespread implementation of neonatal bloodspot screening (NBS) for CF has led to early diagnosis and better outcomes for children with CF is several countries. The rationale for NBS for CF is well established, and there is robust evidence to support this strategy. The goal of CF NBS is to achieve an early CF diagnosis in asymptomatic infants so that comprehensive medical and psychosocial therapies can be implemented to prevent or delay the onset of clinical symptoms [2,3,4,5]. The unwanted consequences of the most recent protocols of NBS for CF (i.e., inclusion of *CFTR* variants’ studies) include the identification of carrier status [6] and the emergence of a cohort of infants with positive NBS test results but an inconclusive diagnosis [7,8,9]. These infants, labelled in the first months of life as CF transmembrane conductance regulator-related metabolic syndrome (CRMS)/CF screen–positive, inconclusive diagnosis (CFSPID) [7], remain healthy in most cases but may receive a CF diagnosis later due to a positive sweat test (ST) or a re-classification of *CFTR* variants as CF, possibly leading to the development of mild clinical CF features [8,9,10,11,12,13].

Currently, owing to improvements in the knowledge of CRMS/CFSPID, several cohorts have been described, and predictors of disease evolution have been hypothesised [8,9,13,14]. Furthermore, the European CF Society Neonatal Screening Working Group has provided guidance on clinical management, highlighting the importance of clear communication to minimise unnecessary and preventable stress and anxiety [8].

Nevertheless, little is known about the psychological impact of the CRMS/CFSPID designation. Johnson et al. conducted five semi-structured interviews with a small number of parents whose children received a CRMS/CFSPID label [15]. This label caused parental distress, which began with the first communication of the results and endured over time, in contrast with parents who receive carrier results, where anxiety dissipates after professional information is obtained [15,16]. In this cohort, this aspect led parents to feel that their child had CF. No data are available on larger case series; moreover, there is no study that compares the impact on the families of a CF infant diagnosed following a positive NBS.

This prospective qualitative and quantitative study compared the psychological impact on parents of children, of Italian descent, with CRMS/CFSPID and CF.

## 2. Materials and Methods

This multicentre study was funded by the Italian Cystic Fibrosis Research Foundation (FFC#24/2020) to evaluate the psychological impact on the parents of CRMS/CFSPID infants. We compared them to parents of CF patients born in 2019–2020 and followed them at four Italian CF centres (Florence, Naples, Ancone, and Brescia). This choice was to compare two groups of the same age and who had recently received the communication of CRMS/CFSPID label or CF diagnosis.

Infants were classified as CRMS/CFSPID if they had a positive NBS result plus: (1) sweat chloride concentration (SCC) < 30 mmol/L and 2 *CFTR* gene variants, at least one of which had unclear phenotypic consequences; or (2) SCC 30–59 mmol/L and 1 or 0 CF-causing variants [7,17]. Patients were defined as having CF if they were NBS-positive with a raised SCC (≥60 mmol/L) and/or two CF-causing variants of the *CFTR* gene [18].

Preliminary anamnestic data were collected, which included socio-demographic data as well as the presence of traumatic or significant events that occurred in the previous 12 months, such as the death of a loved one, separation/divorce, COVID-19 pandemic-related problems, economic problems, changes in home, city, or work, and serious injury/illness. Finally, we asked whether families had ever used any psychological support services provided by the CF Centre and, if so, in what ways and at what times.

The interviews were conducted with both parents in the presence of a psychologist, specialised in CF, belonging to the team of CF centres included in the study.

The study was approved by the Ethical Committee of the CF Coordinator Centre (Florence, Italy, Ethics Clearance number 235/2020, on 29 September 2020). Informed consent was obtained from each parent.

### 2.1. Quantitative Tools

The participants were administered three quantitative survey tools, with high validity and reliability: the Impact of Event Scale-Revised (IES-R) [19], the Patient Health Questionnaire (PHQ-9) [20], and the Generalized Anxiety Disorder (GAD-7) scale [21].

The IES-R is a self-assessment scale that estimates the subjective distress caused by traumatic events and is comprised of 22 questions. Respondents are asked to identify a specific stressful life event and then indicate how distressed or bothered they have been for the last seven days, in relation to each of these listed difficulties, on a 5-point scale ranging from 1 = ‘not at all’ to 5 = ‘extremely’; the scale provides an average total score ranging from 0 to 88 and shows the results of three subscales: avoidance, intrusiveness, and hyperarousal [19].

The PHQ-9 is a questionnaire comprising nine questions that investigate the depressive symptoms found in the last two weeks, referring to the criteria of the diagnostic and statistical manual (DSM)-IV and also included in the DSM-5, which are evaluated with a 5-point scale ranging from 0 = ‘never’ to 4 = ‘almost every day’. There is a final question, whose answer does not contribute to the total score, but estimates the overall functional impairment of depressive manifestations in the patient’s daily work and interpersonal life. The total score ranges from 0 to 27. Values between 5 and 9 indicate the presence of subthreshold depression, whereas those above the cut-off of 10 highlight depressive states of clinical relevance, with three different levels of severity, depending on the score [20].

The GAD-7 is a self-assessment scale validated to screen for generalised anxiety disorder and to estimate the severity of symptoms present in the last two weeks by assigning an attribute score on the 4-point scale ranging from 0 = ‘never’ to 3 = ‘almost every day’. Total values ranging from 0 to 4 describe minimal or no symptoms, the values between 5 and 9 show mild symptoms, those between 10 and 14 show moderate symptoms, and those between 15 and 21 show severe symptoms [21].

These tools are freely available in all major world languages (https://www.phqscreeners.com/ (accessed on 01 October 2020).

### 2.2. Qualitative Tools

Psychological adaptation to the CRMS/CFSPID label (interview schedule) is a qualitative tool specific to investigate the experiences of parents of CRMS/CFSPID children [15]. The Italian translation was developed, after direct contact with the author, using a back-translation procedure by two independent translators. Discrepancies emerging from this procedure were discussed until they reached an agreement on a common version.

The tool involved conducting semi-structured interviews that examined how people made sense of their life experiences. Through the interview, parents were asked to reflect on theirs and their child’s personal histories and to share their thoughts, emotions, and memories evoked by the diagnosis communication now and those presented previously. The areas investigated were grouped into macro categories: impact of diagnosis communication on identity, role and parental experience, the image of their child, description of relationships and perceived closeness of family and acquaintances, sharing the diagnosis with others and with the child, representation of healthcare professionals involved, perception of the child’s health status, future information, and beliefs.

The questions were open-ended, with flexible prompts to obtain more details. They were complemented by drawing activities to help parents consider abstract concepts more concretely. This included a series of concentric circles to allow parents to communicate how close they perceived the people who were involved in the communication (family members, acquaintances, and health professionals), or drawings to help position the perception of the child’s health status on a continuum of health and disease.

The interviews were conducted in the hospital by the ward psychologist with knowledge of CF and lasted 20–40 min; they were registered using pseudonyms to remove identifiers and respect anonymity. Subsequently, the audio recordings were transcribed verbatim and analysed to identify and group them according to the aforementioned macro areas.

### 2.3. Statistics

Descriptive statistics for quantitative variables were obtained using normal distribution tests. After checking the homogeneity of the variances using appropriate statistical tests, comparisons between independent samples were performed using a two-tailed student’s *t*-test for the equality of the means. A Chi-Squared test was used to determine the independence of the two categorical variables. The level of statistical significance was expressed as a *p*-value, which was considered statistically significant if it was less than 0.05.

Research manuscripts reporting large datasets that are deposited in a publicly available database should specify where the data have been deposited and provide the relevant accession numbers. If the accession numbers have not yet been obtained at the time of submission, it must be stated that they will be provided during review. They must be provided prior to publication.

## 3. Results

Sixteen families of CRMS/CFSPID participants (mean age at the time of interview: 18 months; range 9–23 months) and 16 families of CF patients diagnosed with positive NBS (mean age at the time of interview: 17 months; range: 8–24 months) were enrolled. The two groups did not differ significantly in terms of mean age at the time of the interview.

The families (eight per centre, all Italian nationality) were chosen based on the date of birth of the child, in a sequential manner, according to the previously scheduled visit at the centre.

### 3.1. Quantitative Tools: GAD-7, PHQ-9, IES-R

The GAD-7 results highlighted a mild elevation of anxiety in both samples, with no significant differences between the two groups of parents.

The PHQ-9 indicated the presence of values relating to subthreshold depressive symptoms in both groups.

The IES-R test presents a high score on the three subscales: avoidance, thought intrusiveness, and hyperarousal, indicating a possible post-traumatic stress disorder as defined by the DSM-5. There were no statistically significant differences between the two groups.

All the results from GAD-7, PHQ-9, IES-R in CRMS/CFSPID, and CF patients’ parents are reported in Table 1.

The results related to the presence of traumatic events in the 12 months preceding the interviews are presented in Table 2.

There were no significant differences between the two parental groups with respect to the presence of traumatic events in the 12 months prior to the CRMS/CFSPID or CF diagnosis in their child (chi-square test = ns).

Parents of children with CF reported previous experiences of a psychological path in most cases. Regarding the possibility of availing it during the post-diagnosis process, they reported the need (almost in all cases) to undertake it within the reference centre and preferably in individual and not group sessions (Table 3).

CRMS/CFSPID parents reported less prior psychological support experience. With regard to its future activation, it is expected that future psychological support programs may take place at the centre (even to a lesser extent in the CF group), without highlighting a particular preference between individual and/or group modalities (Table 3).

### 3.2. Qualitative Tools: Interview

#### 3.2.1. Diagnosis

Based on the procedures of involved centres, the following methods of communicating positive NBS outcomes have been identified:I.Registered mail sent by the screening laboratory or the birth hospital, which suggests they contact the reference centre following the result of the screening;II.The centre, informed by the screening laboratory or birth point, makes a phone call to the family requesting them to report by appointment at the facility following the result of the screening.

The results indicate the need (or the preference) for families (both groups) to receive communication through direct contact with the CF centre staff to experience emotional containment and answer any doubts.

Furthermore, the families (both groups) considered the timeliness of the centre in taking charge to be important, indicating shorter waiting times between the first contact and communication in the structure.

Families (both groups) positively highlighted the experience of meeting with the multidisciplinary team at the time of the communication in the facility: doctors, psychologist, dietician, and physiotherapist.

#### 3.2.2. Parenting Experience: Before/After Stick Person Drawing Test

In the CF group, the recurring words before the diagnosis had a more positive connotation and were more oriented towards the future: joy, love, presence, happiness, giving their best, and having other children. However, following the diagnosis, the most frequent words were attention, fear, commitment, responsibility, presence, love, and sadness (Table 4).

In the CRMS/CFSPID group, prior to diagnosis, parenting was described in terms of love, adventure, emotions, and happiness. In contrast, following the diagnosis, a more pessimistic tone emerged: commitment, fear, helplessness, and attention (Table 4).

The results indicated a negative psychological impact on the parents of both groups after screening, with moderate incisiveness in the CF group.

### 3.3. Child Representation: Before/After Stick Person Drawing Test

In both groups, parents tended to imagine their child as close to normal before diagnosis; however, this representation tended to have more negative connotations following the diagnosis (Table 5).

### 3.4. Informing the Child

At the time of the first communication, CF parents consider when and how they will be able to share information with their children, imagining doing so during adolescence and, if necessary, sooner.

Parents of children with CRMS/CFSPID tend to take their time deciding whether to share information about their health status with their children, in the hopes that an inconclusive diagnosis may prevent the need for it.

### 3.5. Healthcare Professionals: Test Drawing a Concentric Diagram

In both groups, parents reported low primary care paediatrician involvement (assigned individually to each child in accordance with Italian law), relying mainly on referral centre specialists (Table 6).

### 3.6. Healthy/Sick Child: Disease Continuum Drawing Test

CF parents tend to rate their child’s health as normal or slightly ill.

Parents of children with CRMS/CFSPID believe their child to be healthy and consider a possible progression of CF diagnosis not worrying.

### 3.7. Beliefs and Impressions

In both groups, the moment of communication following the NBS results was considered to be of great importance. Parents’ experiences indicate a lower traumatic impact if communication occurs with direct personal contact. Both groups considered the presence of a psychologist essential from their first visit to the centre.

This section may be divided by subheadings. It should provide a concise and precise description of the experimental results and their interpretation, as well as the experimental conclusions that can be drawn.

## 4. Discussion

This is the first study comparing the psychological impact on parents of infants born in the same period, labelled as CRMS/CFSPID, or diagnosed as CF after a positive NBS. The improvement of NBS programmes for CF has resulted in diagnostic uncertainty regarding this disease. Although this allows for an observation and, if needed, an early intervention, it increases these families’ emotional load, which must be considered and supported [22,23,24].

Our data highlight a similar level of distress both in the families of infants with inconclusive diagnoses and in those of CF infants, despite the clear clinical differences between the two groups: GAD-7 highlights a mild elevation of anxiety in both samples, and PHQ-9 reveals depressive symptoms in both groups.

Two aspects were found to be important in reducing distress: providing early and empathically clear information. The impact of the diagnosis is pervasive and involves long-term family plans, primarily the planning of new pregnancies in both groups [25]. Furthermore, NBS is not explicitly requested in the presence of symptoms or signs, and this involves communication of the diagnosis, which is experienced as intrusive [22,23,24,25,26,27]. Therefore, it is crucial to build a therapeutic alliance with family members right from the communication of screening results.

It is necessary to reflect on the methods of communicating diagnoses (both conclusive and unclear) and follow-up elements capable of modulating the emotional load and therapeutic concordance [26,27]. Further investigation is needed to determine the best protocol for communicating diagnoses (i.e., SPIKES). It is essential that these families maintain trust in the team and the follow-up process, especially in a situation inconsistent with the traditional medical model, because the CRMS/CFSPID label subverts shared ideas of health and disease through the communication of a medical result, which does not provide a diagnostic definition and leaves parents in uncertainty. This does not correspond to the image of a baby who appears completely healthy. Moreover, in these cases, the medical results did not result in a confirmed CF diagnosis or its clear exclusion.

After NBS, both CF and CRMS/CFSPID labels bring the disease to the family’s attention by activating representations of health and illness as social and personal constructs [27]. At this level, a strictly personal representation of the disease reflects one’s own experience and the affective world, as already highlighted (parent reliving his grief) [15].

Unfortunately, the different procedures of NBS are often not well known or understood by the general population, and parents do not always receive complete information on this topic during pregnancy.

In this study, a different definition of parental experience was highlighted before and after the communication of a certain or inconclusive diagnosis (Table 4), in which parental experience was connoted with more negative tones after the diagnosis, in both groups.

The results of our study highlight specific aspects:-The high emotional impact of the diagnosis of both CF and CRMS/CFSPID labels can be understood in light of the intrusiveness of the screening procedures and successive steps in case of positivity;-The scale of avoidance as a coping strategy appeared to be high in both groups, particularly in the group with a certain diagnosis of CF;-The depression and anxiety that were found to be higher, although not significantly, in the CF group may correspond to the bereavement that these families face in terms of: (a) family plans and planning of further pregnancies; (b) the imaginary/dreamed child; (c) one’s ability or competence to care for the child; (d) a life-changing event; and (e) uncertainty (the time between the result of the NBS and the diagnosis must be minimised);-Hyperarousal may be a reported symptom of stress resulting from the period of maximum uncertainty after the NBS and the incorrect method of communicating NBS procedures and their implications.

Furthermore, the perception of the severity of the disease differs in the two groups. Although CRMS/CFSPID families seem to have higher optimism about the progression of the disease and a greater tendency to strategies oriented towards avoidance and denial, the CF group families tend to normalise the perception of the gravity and improve the perception of health over time. This aspect in the CF group could reflect cognitive adaptation and aligns with Perobelli et al.’s study [25], which revealed an improvement in the field of health perception over time. There is a tendency to think that not communicating the diagnosis of the disease or communicating it partially may protect one’s relatives. Nevertheless, it is widely accepted that the lack of information is deleterious and does not guard children from further distress and is not what they themselves want [28].

By contrast, the CRMS/CFSPID group tend to underestimate the risk of disease progression.

Moreover, the physicians’ communication about the future of an infant diagnosed with CRMS/CFSPID may lead to parental optimism regarding their child’s health status. Additionally, in about 10% of Italian patients, CRMS/CFSPID will evolve into CF [9,29,30]. In our opinion, underestimating the risk of developing CF may not be a negative factor but a protective attitude that positively saves the vision of one’s child as a healthy child, overshadowing the possibility of disease.

Finally, the limited involvement of the primary paediatrician was highlighted. The families of the CRMS/CFSPID participants preferred to maintain contact with the CF centre and often did not inform the primary paediatrician about the child’s health. This aspect is worrying, considering that in asymptomatic CRMS/CFSPID over six years of age, discharge from the CF centre and subsequent territorial care could be hypothesised [8].

Although this is the first study comparing the psychological impact on families of CRMS/CFSPID subjects and CF patients diagnosed by NBS, the study has some important limitations. A control group is missing. Furthermore, there are no significant differences between the two groups, probably also due to the small number of children enrolled. For these reasons we cannot provide definitive conclusions and further studies are needed in this area.

## 5. Conclusions

In conclusion, our results highlight a significant negative emotional impact at first communication in both the CF and CRMS/CFSPID groups. Over time in the CF group, there is a tendency towards normalisation of the perception of gravity. The CRMS/CFSPID group underestimate the probability of disease progression.

Special attention must, therefore, be paid to the information given, the communication methods, the accompaniment of families in the follow-up, and elaboration of the emotional impact of the outcome of NBS. Greater early involvement of the primary paediatrician is desirable given the possibility of continuing the follow-up outside the CF centre in asymptomatic children.

## Figures and Tables

**Table 1 children-10-00177-t001:** Cumulative results of The GAD-7, PHQ-9, and IES-R tools in parents of children with CRMS/CFSPID vs parents of children with CF.

Test	GAD-7	PHQ-9		IES-R			
Sub-Scale	GAD-7 M ± ds	PHQ-9 Items M ± ds	PHQ-9 Question M ± ds	Avoidance scale M ± ds	Intrusion scale M ± ds	Hyperarousal scale M ± ds	IES-R Total M ± ds
CRMS/ CFSPID parents	5.46 ± 3.78	4.20 ± 4.20	0.41 ± 0.63	1.80 ± 0.66	1.90 ± 1.09	1.80 ± 1.05	4.20 ± 2.82
CF parents	7.30 ± 4.46	5.10 ± 4.99	0.60 ± 1.07	2.10 ± 0.37	2.10 ± 1.06	2.40 ± 1.39	5.20 ± 2.57
CRMS/ CFSPID vs. CF *p* value	ns	ns	ns	ns	ns	ns	ns

Abbreviations: CF, Cystic Fibrosis; CRMS, *CFTR*-related metabolic syndrome; CFSPID, cystic fibrosis screen-positive, inconclusive diagnosis; IES-R, Impact of Event Scale-Revised; PHQ-9, Patient Health Questionnaire; GAD-7, Generalized Anxiety Disorder; ns, not significant.

**Table 2 children-10-00177-t002:** Traumatic events in the 12 Months preceding the interview in CRMS/CFSPID and CF patient parents.

Traumatic Events	Frequency N (%)	
	CRMS/CFSPID	CF
Death of a loved one	1 (6.2%)	4 (25%)
Separation/divorce	0 (0%)	2 (12.5%)
Coronavirus emergency	8 (50%)	13 (81.2%)
Economic problems	3 (18.7%)	2 (12.5%)
Change of residence	1 (6.2%)	1 (6.2%)
Change city	1 (6.2%)	0 (0%)
Severe injury/illness	1 (6.2%)	0 (0%)
Other	1 (6.2%)	0 (0%)

Abbreviations: CF, cystic fibrosis; CRMS, *CFTR*-related metabolic syndrome; CFSPID, cystic fibrosis screen-positive, inconclusive diagnosis.

**Table 3 children-10-00177-t003:** Previous psychological support experience in parents of children with CRMS/CFSPID and CF.

	Past Experience of Psychological Support	Future Expectations on the Psychological Service Offered by the Centre				
	Yes	Activation	Providing Facility		Type	
		Yes	Centre	Private	Single	Groups
CF N (%)	11 (68.8)	15 (93.8)	12 (75.0)	3 (18.8)	14 (87.5)	1 (6.2)
CRMS/CFSPID N (%)	5 (31.2)	15 (93.8)	9 (56.3)	6 (37.5)	8 (50.0)	7 (43.8)
CF vs. CRMS/CFSPID *p* value	<0.05	ns	ns		<0.05	

Abbreviations: CF, Cystic Fibrosis; CRMS, *CFTR*-related metabolic syndrome; CFSPID, Cystic Fibrosis screen-positive, inconclusive diagnosis; ns, not significant.

**Table 4 children-10-00177-t004:** Recurring words before and after CRMS/CFSPID label and CF diagnosis.

	Parenting Experience			
	**Before Diagnosis**	**N (%)**	**After Diagnosis**	**N (%)**
CF	Joy	6 (37.5%)	Attention	6 (37.5%)
	Love	4 (25.0%)	Fear	7 (43.7%)
	Presence	2 (12.5%)	Commitment	7 (43.7%)
	Happiness	5 (31.2%)	Responsibility	7 (43.7%)
	To give the best	2 (12.5%)	Presence	4 (25.0%)
	Many children	2 (12.5%)	Love	7 (43.7%)
	Freedom	2 (12.5%)	Sadness	2 (12.5%)
	Lightness	2 (12.5%)	Distress	2 (12.5%)
	Responsibility	4 (25.0%)	Treatment	2 (12.5%)
			Awareness	2 (12.5%)
			Disappointment	2 (12.5%)
CRMS/ CFSPID	Love	6 (37.5%)	Commitment	4 (25.0%)
	Adventures	3 (18.7%)	Fear	4 (25.0%)
	Exciting	5 (31.2%)	Impotence	2 (12.5%)
	Happiness	2 (12.5%)	Attention	2 (12.5%)
	Sweetness	3 (18.7%)	Courage	3 (18.7%)
	Anxiety	6 (37.5%)	Hard	2 (12.5%)
			Insecurity	5 (31.2%)
			Force	8 (50.0%)

Abbreviations: CF, Cystic Fibrosis; CRMS, *CFTR*-related metabolic syndrome; CFSPID, Cystic Fibrosis screen-positive, inconclusive diagnosis.

**Table 5 children-10-00177-t005:** Child representation before and after CF and CRMS/CFSPID diagnosis.

	Child Representation			
	**Before Diagnosis**		**After Diagnosis**	
CF	Freedom	2 (12.5%)	Fragility	9 (56.2%)
	Vivacity	4 (25%.0)	Fighter	6 (37.5%)
	Sweetness	3 (18.7%)	Protection	2 (12.5%)
	Be like others	2 (12.5%)	Sweetness	4 (25.0%)
	Healthy	2 (12.5%)	Dependence	2 (12.5%)
	Love	2 (12.5%)	Vivacity	3 (18.7%)
	Normality	4 (25.0%)	Disease	2 (12.5%)
			Resilient	3 (18.7%)
CRMS/CFSPID	Healthy	9 (56.2%)	Fragility	5 (31.2%)
	Carefree	4 (25.0%)	Attention	5 (31.2%)
	Happiness	2 (12.5%)	Weakness	4 (25.0%)
	Sweetness	9 (56.2%)	Fierce	5 (31.2%)
			Goodness	7 (43.7%)

Abbreviations: CF, Cystic Fibrosis; CRMS, *CFTR*-related metabolic syndrome; CFSPID, cystic fibrosis screen-positive, inconclusive diagnosis.

**Table 6 children-10-00177-t006:** Relations with healthcare for parents of children with CRMS/CFSPID and CF.

	Relations with Healthcare					
	Close Proximity N (%)		Moderate Proximity N (%)		Low Proximity N (%)	
CF	Paediatricians of CF centre	14 (87.5%)	Psychologist	9 (56.2%)	Primary paediatrician	13 (81.2%)
	Nutritionist	3 (18.7%)	Nurses	5 (31.2%)	Nurses	3 (18.7%)
	Psychologist	6 (37.5%)	Paediatricians of CF centre	2 (12.5%)		
CMRS/ CFSPID	Geneticist	6 (37.5%)	Nurses	3 (18.7%)	Primary paediatrician	6 (37.5%)
	Paediatricians of CF centre	7 (43.7%)	Paediatricians of CF centre	3 (18.7%)	Psychologist	3 (18.7%)
	Nurses	3 (18.7%)	Primary paediatrician	2 (12.5%)		

Abbreviations: CF, Cystic Fibrosis; CRMS, *CFTR*-related metabolic syndrome; CFSPID, cystic fibrosis screen-positive, inconclusive diagnosis.

## Data Availability

Data available on request due to restrictions eg privacy or ethical.

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
