# Peer review of "A Multicentre Italian Study on the Psychological Impact of an Inconclusive Cystic Fibrosis Diagnosis after Positive Neonatal Screening"

_children, 2023, doi:10.3390/children10020177_

Round 1

Reviewer 1 Report

Overall comment:

The article is interesting but needs several adjustments to clarify some of its contents.

Abstract: Materials and Methods and tables, and Results:

The qualitative categories for analysis need to be aligned throughout the article. Different phrases and terms were used to describe the same categories.

Section Materials and Methods:

Reason for sample selection needs clarification;

Lines 108-111: The PHQ-9 point scale needs to be clarified. The original is a 4-point scale;

Lines 158-172: This seems to be text copied from the authors' guidelines (?).

Results:

Tables should be included first, followed by their interpretation.

Lines 275-277: This seems to be text copied from the authors' guidelines (?).

Discussion:

Line 279. Authors refer to 'a large sample size'. Considering that there were 32 parents involved, authors might consider rephrasing.

Author Response

The article is interesting but needs several adjustments to clarify some of its contents.

Re: We thank you for carefully evaluating our paper and for your favourable comments.

Abstract: Materials and Methods and tables, and Results:

The qualitative categories for analysis need to be aligned throughout the article. Different phrases and terms were used to describe the same categories.

Re: We thank the reviewer for the suggestion. We have modified the text and tables accordingly

Section Materials and Methods:

Reason for sample selection needs clarification;

Re: We understand the reviewer's request; we have added (in yellow) the following sentence in methods: “This choice was to compare two groups of the same age and who had recently received the communication of CRMS/CFSPID label or CF diagnosis”.

Lines 108-111: The PHQ-9 point scale needs to be clarified. The original is a 4-point scale;

Re: We have referred to the paper of Kroenke, K. and Spitzer, R.L. The PHQ-4 point is an ultra-brief screening scale for anxiety and depression; we are aware that it can replace PHQ-9, but it also lacks some important items.

Lines 158-172: This seems to be text copied from the authors' guidelines (?).

Re: We apologize for this mistake; we have eliminated the non-useful section.

Results:

Tables should be included first, followed by their interpretation.

Re: we have modified the results, always reporting the table first and then the interpretation.

Lines 275-277: This seems to be text copied from the authors' guidelines (?).

Re: In the original file lines 275-277 correspond to the following sentence “however, this representation tended to have more negative connotations following 275 the diagnosis (Table 5).”

Discussion:

Line 279. Authors refer to 'a large sample size'. Considering that there were 32 parents involved, authors might consider rephrasing.

Re: We agree. We have changed the sentence. Regardless of the sample size, our study is the first that compare the psychological impact on CRMS/CFSPID and CF children diagnosed with positive newborn screening.

Reviewer 2 Report

The study ia an important one. However, the main aim of the research has not depicted by the results. Imoact of independent variable on the dependent variable has not adequately addressed. QUalitative data analysis also quite hard to understand. 

Author Response

Reviewer 2

The study ia an important one. However, the main aim of the research has not depicted by the results. Impact of independent variable on the dependent variable has not adequately addressed. Qualitative data analysis also quite hard to understand.

Re: We fully agree with the reviewer. This is the first study comparing the psychological impact on families of CRMS/CFSPID subjects and CF patients diagnosed by newborn screening and age-matched. Nonetheless, the study has some important limitations that we have better specified in the discussion. A control group is missing. A multivariate analysis was not performed due to the small number of the sample.  

Round 2

Reviewer 2 Report

The manuscript has developed according to the feedback. However, please consider the commented places for the improvement of this manuscript. 

Author Response

1. Mentioning the numerical values make the abstract more complicated. You   can mention that in the text. But not in the abstract.

R. As suggested by the reviewer, we have deleted the numerical values from the abstract.

2. This toll should be explained in a paragraph. Authors and years also should be highlighted.

R. We anticipated the explanation in the paragraph and then quoted the table. The other required information, such as the age of the two groups of subjects, are already reported in the text.

3. Better to convert the all the tables according to APA style.

R. As suggested by the reviewer we changed all the tables to APA style

4. A Conclusion can be integrated.

R. We have inserted the paragraph of conclusions